# A Photoexcited Switchable Dual-Function Metamaterial Absorber for Sensing and Wideband Absorption at THz Band

**DOI:** 10.3390/nano12142375

**Published:** 2022-07-11

**Authors:** Liansheng Wang, Dongyan Xia, Quanhong Fu, Yuan Wang, Xueyong Ding

**Affiliations:** 1School of Science and Technology, Sanya University, Sanya 572022, China; yuanwang@sanyau.edu.cn (Y.W.); xueyongding@sanyau.edu.cn (X.D.); 2School of Finance and Economics, Sanya University, Sanya 572022, China; dongyanxia@sanyau.edu.cn; 3School of Physical Science and Technology, Northwestern Polytechnical University, Xi’an 710129, China; fuquanhong@nwpu.edu.cn

**Keywords:** THz band, sensing, wideband absorption, metamaterial absorber

## Abstract

Based on the tunable conductivity of silicon as a function of incident pump power, a photoexcited switchable dual-function metamaterial absorber for sensing and wideband absorption at the THz band is designed in this paper. The absorber has an absorption peak at 2.08 THz with the absorption up to 99.6% when the conductivity of silicon is 150 Sm^−1^, which can be used for sensing. The refractive index sensitivity of the absorption peak is up to 456 GHz/RIU. A wideband absorption is generated from 3.4 THz to 4.5 THz with the bandwidth of 1.1 THz as the conductivity σ_si_ = 12,000 Sm^−1^. The generation mechanism of the sensing absorption peak and wideband absorption is explained by monitoring the surface current, electric, and magnetic field distribution at some absorption frequencies. It has the advantages of being simple and having a high sensitivity, and wideband absorption with wide application prospects on terahertz communication, electromagnetic stealth, and biochemical detection.

## 1. Introduction

A metamaterial is a kind of periodic artificial structure that is composed of sub-wavelength resonant units. Its electromagnetic properties can be artificially controlled by designing its unit cell and it holds important application values in electromagnetic stealth, plate focusing, polarization conversion, and direction modulation [1,2,3,4,5,6,7,8,9,10,11]. The metamaterial absorber is an important research aspect of a metamaterial. The perfect absorption of incident electromagnetic waves can be realized by optimizing the unit cell of a metamaterial. With the deepening of research, metamaterial absorber has made great progress in achieving polarization-insensitivity and wide incident angle response at microwave, terahertz, and infrared band [12,13,14,15,16,17,18,19]. Compared with single-frequency and multi-frequency metamaterial absorber, wideband metamaterial absorber has broader application prospects.

The frequency of electromagnetic waves at the THz band is 0.1–10 THz. The electromagnetic wave at this band has the advantages of high transmittance, wider frequency bandwidth, and a higher signal-to-noise ratio. It is widely used on radar, remote sensing, environmental monitoring, and wideband communication. Therefore, it is of great significance to study the wideband metamaterial absorber at the THz band. Laminated structure and loaded resistance film are important methods to realize wideband absorption of metamaterial absorber at THz band. In this aspect, Qiu Y Q et al. designed a wideband terahertz metamaterial absorber based on a symmetrical L-shaped metal resonator [20]; its absorption exceeds 90% from 0.457 THz to 1 THz. Based on the phase transition property of vanadium dioxide, Zhang M et al. proposed a dual-function metamaterial absorber with a switchable function of wideband absorption and multi-band absorption at the THz band [21]. When the VO_2_ acts as metal, the metamaterial absorber obtains wideband absorption with the absorption of more than 90% from 3.25 THz to 7.08 THz. Pan H et al. proposed a thermally tunable ultra-wideband polarization-insensitive metamaterial absorber at the THz band [22]. At the temperature of 340 K, the absorption of the metamaterial absorber is more than 90% at the range of 2.38–21.13 THz with a relative band of 159.5%. Quader S et al. designed a graphene-based ultra-wideband tunable metamaterial absorber at the THz band [23], its absorption exceeds 90% from 0.1 THz to 3.1 THz and from 6.25 THz to 8.55 THz with the relative bandwidth of 187.5% and 31%, respectively.

The wideband metamaterial absorber at the THz band has important application values on terahertz electromagnetic stealth and communication, while the narrowband terahertz metamaterial absorber has broad application prospects in terahertz biosensors and other fields. In the research on narrowband terahertz metamaterial absorbers, Pang H Z and his team proposed a dual-band terahertz metamaterial absorber [24]. Its absorption is more than 99% at the resonant frequency of 0.387 THz and 0.694 THz with the quality factor *Q* = 28.1 and 29.3, respectively, and the refractive index sensitivity is 39.5 and 85 GHz/RIU, respectively. Wang J L et al. designed a terahertz metamaterial absorber based on an I-shaped resonant structure [25]. It has an extremely narrow absorption peak with absorption of 99.5% at 0.523 THz with a quality factor of 37. The team of Li Y R proposed a terahertz metamaterial absorber based on metal rings with different radii loaded on the dielectric layer [26]. It has two absorption peaks to work as a biosensor at 2.335 THz and 4.215 THz with an absorption up to 99.99%. Wang X et al. designed a terahertz high-sensitivity refractive index sensor based on a metamaterial absorber, which is composed of the three-dimensional split resonant ring array and microfluidic channels [27], and the refractive index sensitivity reached up to 379 GHz/RIU.

The above-mentioned wideband and narrowband metamaterial absorbers at the THz band have a single function, while the multi-function terahertz metamaterial absorber has wider application prospects in practice. In this paper, we designed a photoexcited switchable dual-function metamaterial absorber for sensing and wideband absorption at the THz band based on the tunable conductivity of silicon as a function of incident pump power. It has the advantages of simple structure, high sensitivity, wideband absorption, and switchable dual-function with wide application prospects on terahertz communication, electromagnetic stealth, and biochemical detection

## 2. Model Design

Our designed photoexcited switchable dual-function metamaterial absorber for sensing and wideband absorption at the THz band is shown in Figure 1. It is composed of four layers: split gold ring, silicon disk, polyimide, and gold substrate along the negative direction of the *z*-axis. The optimized structural dimension parameters are: *a* = *b* = 15 μm, *r*_2_ = 6.75 μm, *r*_1_ = 6 μm, *c* = 1 μm, *d* = 0.5 μm, *t*_1_ = 0.1 μm, *t*_2_ = 1 um, *t*_3_ = 6 μm, *t*_4_ = 1 μm. The conductivity of gold *σ* = 4.56×10^7^s/m [28]. The conductivity of silicon is a function of incident pump power. In our design, the conductivity of silicon is set as *σ*_si_ = 150 Sm^−1^ and 12,000 Sm^−1^, respectively with the permittivity *ε*_r_ = 11.7 [28]. The permittivity of polyimide *ε*_r_ = 3.5, and its tangent of loss angle tan *δ* = 0.0027 [29].

The electromagnetic simulation software Microwave Studio CST is used to design and simulate the unit cell shown in Figure 1. During the simulation process, the boundary condition of the *x* and *y* directions is set as unit cell and the *z*-direction is set as open. The incident wave is set as a plane wave that propagates along the negative direction of the z-axis, and the electric field and magnetic field of the incident electromagnetic wave are along the *x*-axis and *y*-axis, respectively. The frequency-domain solver is used to simulate the electromagnetic parameters of the unit cell.

## 3. Results and Discussion

The absorption of metamaterial absorber with σ_si =_ 150 Sm^−1^ and 12,000 Sm^−1^ is shown in Figure 2. It can be seen from Figure 2 that an absorption peak appears at 2.08 THz, with an absorption up to 99.6%, which can be used for sensing when the conductivity of silicon σ_si_ = 150 Sm^−1^. When the conductivity of silicon σ_si_ = 12,000 Sm^−1^, the absorption of the metamaterial absorber exceeds 90% from 3.4 to 4.5 THz acquiring a bandwidth of 1.1 THz. The above results show that the metamaterial absorber has the photoexcited switchable dual-function for sensing and wideband absorption at the THz band.

In order to study the causes of the absorption peak for sensing and wideband absorption, the absorption of the metamaterial absorber in the case of only having a split gold ring and silicon disk with σ_si_ = 12,000 Sm^−1^ is simulated and calculated. The results are shown in Figure 3. When there is only a split gold ring, a narrow-band absorption peak appears at 2.08 THz, with an absorption of 99.6%, and the absorption of the metamaterial absorber exceeds 90% from 3.78 THz to 5.78 THz, with the bandwidth of 2 THz when there is only a silicon disk. The above results indicate that the absorption peak for sensing and wideband absorption of the metamaterial absorber is generated by the split gold ring and the silicon disk under the action of the incident wave, respectively.

In order to deeply explore the reason behind the formation of the absorption peak for sensing at 2.08 THz, the surface current distribution at 2.08 THz with σ_si_ = 150 S/m is monitored, as shown in Figure 4. The surface current of the metamaterial absorber is mainly concentrated on the split gold ring. It flows along the ring and then forms a current loop, which indicates that the metamaterial absorber generates magnetic resonance under the action of the incident wave. Therefore, the main reason for the absorption peak for sensing at 2.08 THz is the magnetic resonance generated by the metamaterial absorber under the action of the incident wave [30].

For further verification of the mechanism of wideband absorption, the electric and magnetic field distributions of the metamaterial absorber at 3.6 THz, 4 THz, and 4.5 THz with *σ*_si_ = 12,000 Sm^−1^ are monitored, as shown in Figure 5. The electric field and the magnetic field are mainly concentrated on the middle part of the metamaterial absorber, which indicates that the metamaterial absorber has strong electromagnetic resonance under the action of incident waves. The produced electromagnetic resonance results in the strong absorption of incident waves [31]. The superposition of different resonant frequencies leads to wideband absorption.

The above results are obtained at the condition of the incident wave vertically on the metamaterial absorber. In order to study the relationship of the absorption peak for sensing and wideband absorption with the polarization and incident angle of the incident wave, the absorption of the metamaterial absorber under different polarization and incident angles is calculated, as shown in Figure 6, Figure 7 and Figure 8. The absorption at 2.08 THz with *σ*_si_ = 150 Sm^−1^ and wideband absorption from 3.4 to 4.5 THz with σ_si_ = 12,000 Sm^−1^ gradually decreases with an increase in polarization angle. It indicates that the absorption peak for sensing and wideband absorption of the metamaterial absorber are polarization-sensitive due to the non-rotational symmetry of the unit cell. When the conductivity of silicon σ_si_ = 150 Sm^−1^, the absorption at 2.08 THz gradually decreases with an increase in the incident angle at TE mode, but the absorption peak for sensing at 2.08 THz disappears at TM mode. When the conductivity of silicon σ_si_ = 12,000 Sm^−1^, the absorption from 3.4 THz to 4.5 THz gradually decreases with an increase in the incident angle at TE mode, but the wideband absorption from 3.4 THz to 4.5 THz also disappears at TM mode.

The main reason for the gradual decreasing of the absorption with an increase in the incident angle is that the impedance matching degree of the metamaterial absorber with free space gradually decreases with the increasing of the incident angle [32]. The above results show that the absorption peak for sensing and wideband absorption of the metamaterial absorber is sensitive to the incident angle.

Figure 9 and Figure 10 show the absorption of the metamaterial absorber with σ_si_ = 150 Sm^−1^ and σ_si_ = 12,000 Sm^−1^ under different dimensional parameters c and d. The absorption peak for sensing gradually shifts to a higher frequency with the increasing of the dimensional parameter c, but it gradually shifts to a lower frequency with the absorption gradually decreasing when the dimensional parameter d increases. The bandwidth of wideband absorption gradually increases with the increase of dimensional parameter c, but it decreases with the increase of dimensional parameter d.

## 4. The Sensing Property of Metamaterial Absorber

In order to analyze the sensing property of the metamaterial absorber, we placed the object to be measured above the split gold ring and varied its refractive index from 1 to 3 with a thickness of 1 μm. The absorption of the metamaterial absorber under different refractive indexes of the object with σ_si_ = 150 Sm^−1^ is shown in Figure 11. The absorption peak for sensing at 2.08 THz gradually moves to a lower frequency with the increase of the refractive index. The top split gold ring of the metamaterial absorber may be equivalent to an *LC* resonant circuit, and the resonant frequency of the *LC* resonance circuit is $f\propto 1/(2\pi \sqrt{LC})$. The capacitance *C* at the slot of the split gold ring gradually increases with the increase of the refractive index, which leads to the absorption peak for sensing moving to a lower frequency. According to the calculation formula of sensitivity S=Δf/Δn, the refractive index sensitivity of the metamaterial absorber is up to 456 GHz/RIU. At the same time, the quality factor *Q* of the metamaterial absorber at 2.08 THz is up to 6.5 according to the calculation formula of quality factor *Q* = *f/FWHM* (*f* is the central resonant frequency, and *FWHM* is the half height width of its corresponding transmission resonant peak). The performance comparison of the absorption peak for sensing at 2.08 THz with other devices from references [24,25,27] is shown in Table 1. We can see from Table 1 that the sensitivity of the metamaterial absorber is very high.

## 5. Summary

We present a photoexcited switchable dual-function metamaterial absorber for sensing and wideband absorption at the THz band based on the tunable conductivity of silicon as a function of incident pump power. An absorption peak for sensing appears at 2.08 THz with absorption up to 99.6% when the conductivity of silicon is 150 Sm^−1^. The refractive index sensitivity of the absorption peak is up to 456 GHz/RIU. The wideband absorption is generated from 3.4 THz to 4.5 THz, providing a bandwidth of 1.1 THz when the conductivity of silicon σ_si_ = 12,000 Sm^−1^. The surface current distribution of the metamaterial absorber at 2.08 THz shows that the absorption peak for sensing is derived from the generated magnetic resonance of the top split gold ring under the action of incident waves. The electric and magnetic field distribution at 3.6 THz, 4 THz, and 4.5 THz show that the wideband absorption originates from the superposition of different resonance frequencies which are produced from the electromagnetic resonance of the metamaterial absorber under the action of the incident wave. Due to the non-rotational symmetry of the unit cell, the absorption property of the metamaterial absorber is sensitive to both polarization and incident angles. It has the advantages of being simple, highly sensitive, and wideband absorption providing application prospects on terahertz communication, electromagnetic stealth, and biochemical detection.

## Figures and Tables

**Figure 1 nanomaterials-12-02375-f001:**
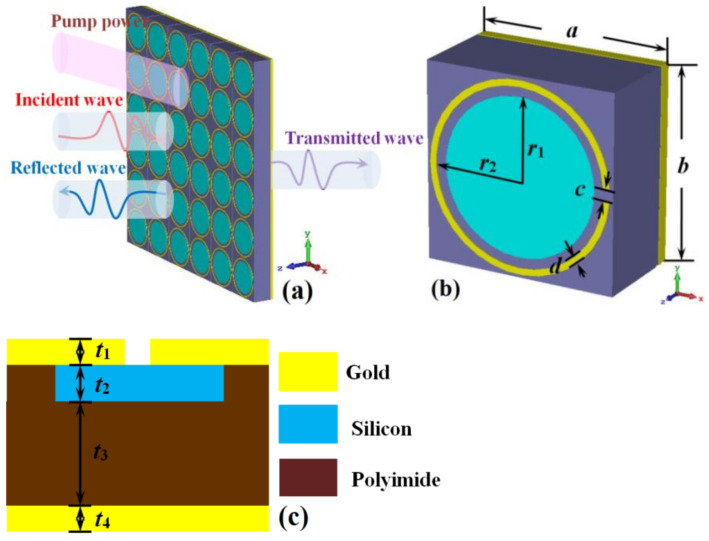
The unit cell of switchable dual-function metamaterial absorber at THz band, (**a**) 6 × 6 arrays; (**b**) perspective view; (**c**) side view.

**Figure 2 nanomaterials-12-02375-f002:**
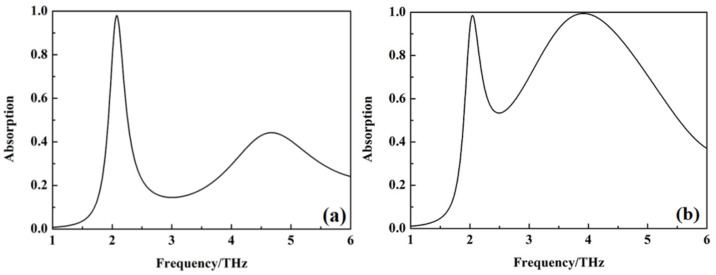
The absorption curve of the metamaterial absorber with different conductivity of silicon, (**a**) σ_si_ = 150 Sm^−1^; (**b**) σ_si_ = 12,000 Sm^−1^.

**Figure 3 nanomaterials-12-02375-f003:**
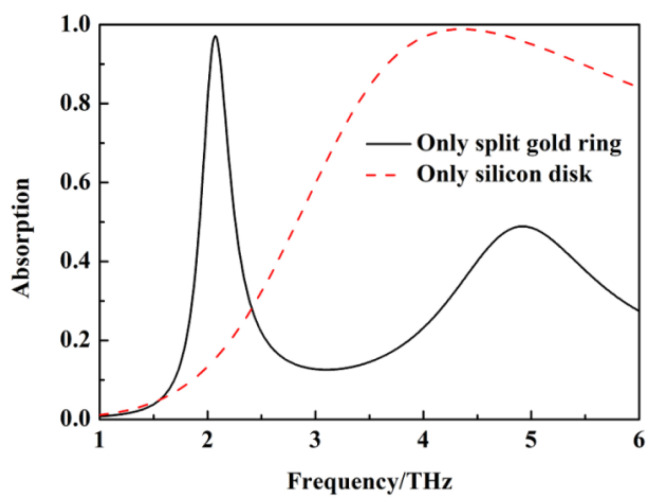
The absorption of metamaterial absorber in the case of only having split gold ring and silicon disk with σ_si_ = 12,000 Sm^−1^.

**Figure 4 nanomaterials-12-02375-f004:**
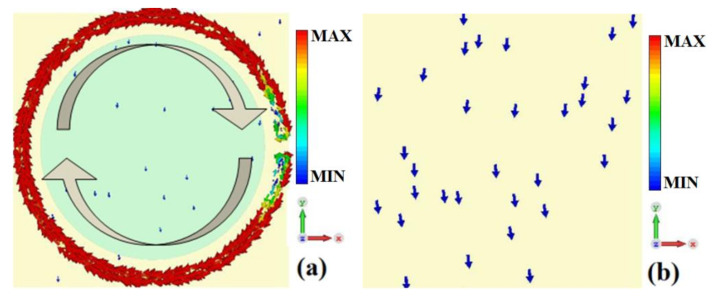
The surface current distribution of metamaterial absorber at 2.08 THz with σ_si_ = 150 Sm^−1^, (**a**) top layer; (**b**) bottom layer.

**Figure 5 nanomaterials-12-02375-f005:**
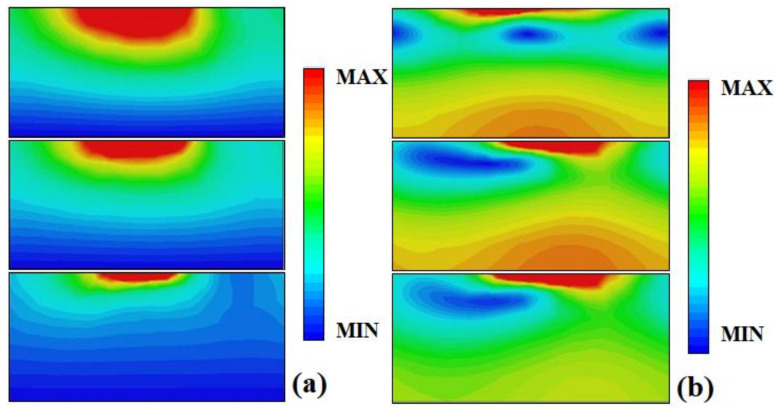
The electric field and magnetic field distribution of metamaterial absorber at 3.6 THz (upper), 4 THz (middle), and 4.5 THz (lower) with σ_si_ = 12,000 Sm^−1^, (**a**) electric field; (**b**) magnetic field.

**Figure 6 nanomaterials-12-02375-f006:**
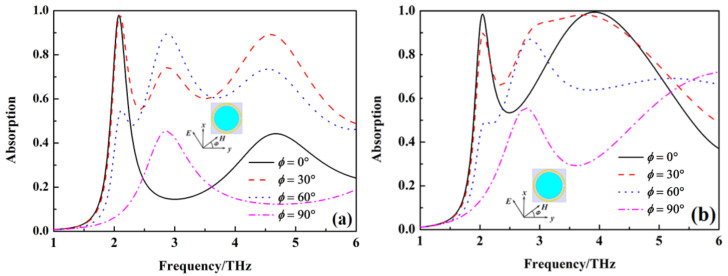
The absorption of metamaterial absorber under different polarization angle, (**a**) σ_si_ = 150 Sm^−1^, (**b**) σ_si_ = 12,000 Sm^−1^.

**Figure 7 nanomaterials-12-02375-f007:**
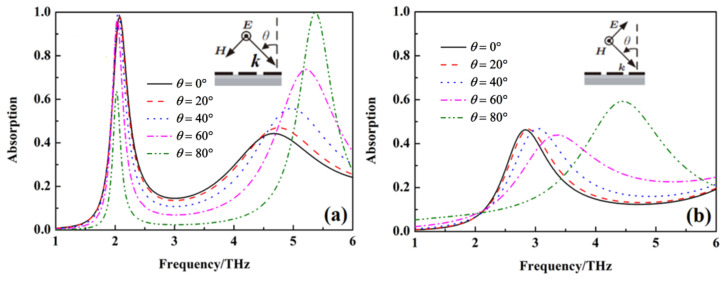
The absorption of metamaterial absorber under different incident angles with σ_si_ = 150 Sm^−1^, (**a**) TE mode, (**b**) TM mode.

**Figure 8 nanomaterials-12-02375-f008:**
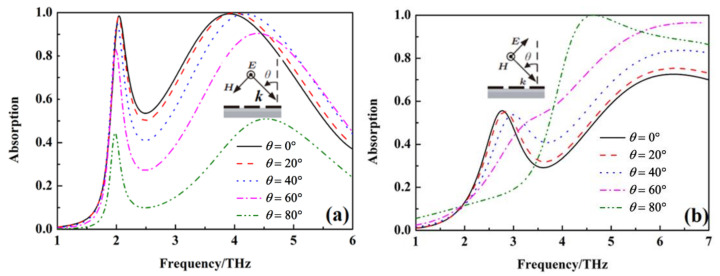
The absorption under different incident angles with σ_si_ = 12,000 Sm^−1^, (**a**) TE mode, (**b**) TM mode.

**Figure 9 nanomaterials-12-02375-f009:**
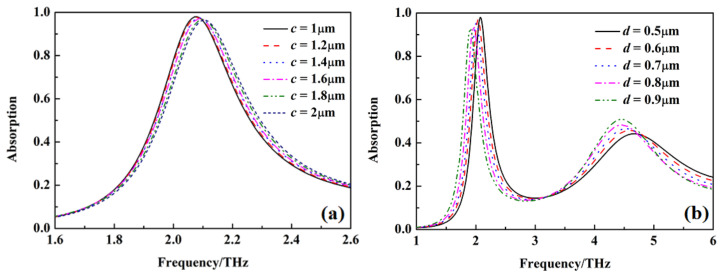
The absorption of metamaterial absorber under different *c* and *d* with σ_si_ = 150 Sm^−1^, (**a**) different *c*, (**b**) different *d*.

**Figure 10 nanomaterials-12-02375-f010:**
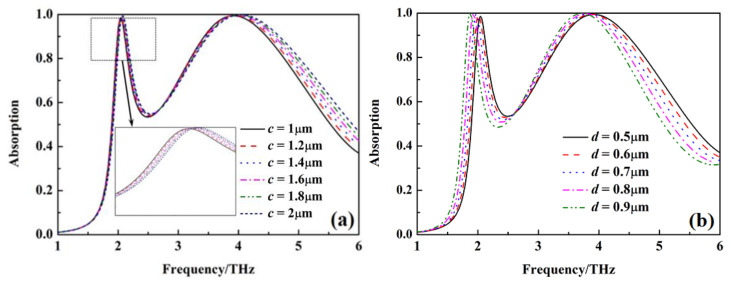
The absorption of metamaterial absorber under different *c* and *d* with σ_si_ = 12,000 Sm^−1^, (**a**) different *c*, (**b**) different *d*.

**Figure 11 nanomaterials-12-02375-f011:**
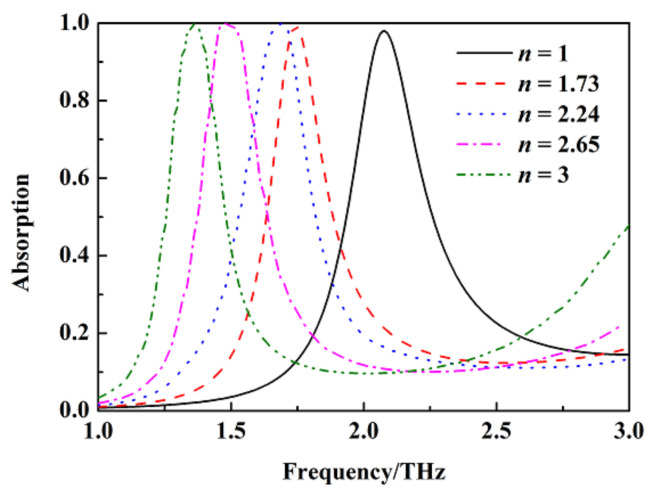
The absorption of the metamaterial absorber under different refractive indexes of the object to be measured with σ_si_ = 150 Sm^−1^.

**Table 1 nanomaterials-12-02375-t001:** The performance comparison of the absorption peak for sensing at 2.08 THz with other devices from references [24,25,27].

Absorber	Absorption Frequency	Absorption	Sensitivity *S*	Quality Factor *Q*
In this paper	2.08 THz	99.6%	456 GHz/RIU	6.5
In Ref. [24]	0.387, 0.694 THz	99%	39.5, 85 GHz/RIU	28.1, 29.3
In Ref. [25]	0.523 THz	99.5%	80.7 GHz/RIU	37
In Ref. [27]	0.79 THz	98.8%	379 GHz/RIU	53

## Data Availability

The data that support the findings of this study are available from the corresponding author upon reasonable request.

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
