# Peer review of "A Photoexcited Switchable Dual-Function Metamaterial Absorber for Sensing and Wideband Absorption at THz Band"

_nanomaterials, 2022, doi:10.3390/nano12142375_

Round 1
Reviewer 1 Report
This study suggests a photoexcited switchable dual-function metamaterial absorber for sensing and wideband absorption at the THz band. This paper is publishable for "Nanomaterials" if the author addresses following comments.
1 The author mentioned that the absorption is up to 99.6% at 2.08 THz absorption peak. The reviewer is wondering if this performance is remarkable. A table showing performances of other devices from references is required.
2 The author suggested a specially designed absorber and mentioned an advantage of high sensitivity. Is there any parameter showing high sensitivity?
3 In the device structure, what is the role of polyimide layer?
4 In figure 2, there are two main peaks: a narrow peak (~2 THz), a broad peak (3~5 THz). It is recommended to define the origin of those peaks in the manuscript.
5 In figure 6, the absorption gradually decreases with an increase in the incident angle. Is there any particular reason?
6 Interestingly, the device structure might be able to be applied to various applications. For a practical application, is it possible to show light detecting behaviors such as IV curve under light illumination or light switching dynamics. That would be useful for researchers studying photodetectors.
Author Response
Dear reviewer,
Thank you very much for your constructive criticisms and valuable comments that undoubtedly improved the quality of our paper.
Please find below our response to the comments of the reviewers and the corresponding changes we made to the manuscript; the changes are highlighted in red.
Yours sincerely,
Dr. Liansheng Wang (On behalf of all authors)

Reviewer 2 Report
The article describes the model of metasurface-based absorber in THz range that combines a sharp absorption peak sensitive to the environment, and a wide absorption band. The spectral response of the absorber can be controlled with the external optical pump, giving some kind of “dual functionality”. The idea looks very interesting. However, I feel the manuscript needs a major revision before it can be published in Nanomaterials since there are a number of inconsistencies. Before I start to list the problems with the manuscript, I’d like to mention that the proposed design can be considered rather continuously-adjustable than switchable, since the parameters of the metasurface is controlled with optical pump: continuous sweep of pump power should lead to continuous transition of the metasurface between “sensing” and “absorbing” functionality.
The main problems I found are as follows:
1. Absorber design description is not consistent with Fig. 1: Si disk with r1 = 6 um and gold ring with r = 6.75 um do not fit to the 7.5x7.5 um square since its diameters (= 2r = 12 um and 13.5 um respectively) are larger than the square side length.
2. The absence of spectral shift of sensing peak under change in gap in Au ring (Fig. 9(a) and Fig. 10(a)) looks inconsistent with author’s statement that this peak arises from the ring resonance (Fig. 3) and, also, with the spectral shift of this peak under change in index of the material inside the gap (Fig. 11). It seems that increase of the gap width “c” should lead to decrease of the capacity C the same way it changes under the decrease in the permittivity of the material in the gap.
3. The most concerns I have is about “switching” of the absorber with the external pump.
First, why authors used exactly those values for Si conductivity?
Next, it will be better to provide the parameters of the pump (power, wavelength) suitable for such conductivity values. Since it rather high, in case of visible light pump (interband absorption in Si) the power should be rather high. Just a raw simple estimations: 150 and 12000 S/m corresponds to roughly 10^16 and 10^18 1/cm^3 of photoexcited charge carriers. With the typical Si interband absorption of about 100 1/cm and photoexcited carriers lifetime of about 1 us this gives about 0.5 and 50 mW of pump power with photon energy 2 eV under conditions that pump spot uniformly illuminate all the absorber surface area. 50 mW of optical power for switching seems to be rather high value that will lead to heating of Si and, consequently, change its properties.
Next, the model assumes the uniform distribution of the pump intensity over the entire 6x6 matrix, because conductivity of all Si disks assumed the same. In practice, it is really hard to make an ideally “flat top” spot of pump light. In real device, the parameters of different disks in the matrix will vary according to the distribution of the pump intensity. It is not clear, is it possible to save the metasurface absorber properties under these conditions.
Also, there are a number of minor corrections I recommend to make the article better:
1. Please, provide references for all material data, like polyimide permittivity and gold conductivity. This values are static values, isn’t it? (can be confused with high-frequency permittivity/conductivity at the frequency of incident THz radiation)
2. Absolute values of surface currents/fields distribution (Fig. 4,5) make no sense while the intensity of incident THz plane wave used in model is not specified. Please, provide this data.
3. Vector field images (Fig. 5) are really hard to understand. It would be better to add some scalar-valued (for example, |E|) cross-sectional image of the field distributions.
Author Response

(The authors gave the same response as above.)

Round 2
Reviewer 2 Report
Dear Authors,
thank you for addressing all my comments.
Unfortunately, my concerns about parameters of optical pump and realistic pump intensity distribution are still not cleared up: the reference you cite only provide the same concept without any details of pump parameters.
However, I suppose, the main idea of the manuscript is the design of the metasurface, and the questions I rise are slightly outside of the article scope. There are no doubts that conductivity of Si can be changed by external illumination, all my comments were formally addressed, so I think the manuscript can be accepted for publication in Nanomaterials in present form.